# Methodological Flaws in Meta-Analyses of Clinical Studies on the Management of Knee Osteoarthritis with Stem Cells: A Systematic Review

**DOI:** 10.3390/cells11060965

**Published:** 2022-03-11

**Authors:** Christoph Schmitz, Christopher Alt, David A. Pearce, John P. Furia, Nicola Maffulli, Eckhard U. Alt

**Affiliations:** 1Chair of Anatomy II, Institute of Anatomy, Faculty of Medicine, Ludwig-Maximilians University, 80336 Munich, Germany; 2Isar Klinikum, General Management, 80331 Munich, Germany; christopher.alt@ingeneron.com (C.A.); ealt@tulane.edu (E.U.A.); 3InGeneron Inc., Houston, TX 77054, USA; 4Department of Pediatrics, Sanford School of Medicine, University of South Dakota, Sioux Falls, SD 57105, USA; david.pearce@sanfordhealth.org; 5Sanford Health, Sioux Falls, SD 57117, USA; 6Sanford Research, Sioux Falls, SD 57105, USA; 7SUN Orthopedics of Evangelical Community Hospital, Lewisburg, PA 17837, USA; jfuria@ptd.net; 8Department of Musculoskeletal Disorders, Faculty of Medicine and Surgery, University of Salerno, 84084 Fisciano, Italy; n.maffulli@qmul.ac.uk; 9Centre for Sports and Exercise Medicine, Barts and the London School of Medicine and Dentistry, Mile End Hospital, Queen Mary University of London, London E1 2AD, UK; 10School of Pharmacy and Bioengineering, Guy Hilton Research Centre, Keele University School of Medicine, Stoke on Trent ST4 7QB, UK; 11Heart and Vascular Institute, Department of Medicine, Tulane University Health Science Center, New Orleans, LA 70112, USA

**Keywords:** meta-analyses, primary knee osteoarthritis, stem cells, systematic review

## Abstract

(1) Background: Conclusions of meta-analyses of clinical studies may substantially influence opinions of prospective patients and stakeholders in healthcare. Nineteen meta-analyses of clinical studies on the management of primary knee osteoarthritis (pkOA) with stem cells, published between January 2020 and July 2021, came to inconsistent conclusions regarding the efficacy of this treatment modality. It is possible that a separate meta-analysis based on an independent, systematic assessment of clinical studies on the management of pkOA with stem cells may reach a different conclusion. (2) Methods: PubMed, Web of Science, and the Cochrane Library were systematically searched for clinical studies and meta-analyses of clinical studies on the management of pkOA with stem cells. All clinical studies and meta-analyses identified were evaluated in detail, as were all sub-analyses included in the meta-analyses. (3) Results: The inconsistent conclusions regarding the efficacy of treating pkOA with stem cells in the 19 assessed meta-analyses were most probably based on substantial differences in literature search strategies among different authors, misconceptions about meta-analyses themselves, and misconceptions about the comparability of different types of stem cells with regard to their safety and regenerative potential. An independent, systematic review of the literature yielded a total of 183 studies, of which 33 were randomized clinical trials, including a total of 6860 patients with pkOA. However, it was not possible to perform a scientifically sound meta-analysis. (4) Conclusions: Clinicians should interpret the results of the 19 assessed meta-analyses of clinical studies on the management of pkOA with stem cells with caution and should be cautious of the conclusions drawn therein. Clinicians and researchers should strive to participate in FDA and/or EMA reviewed and approved clinical trials to provide clinically and statistically valid efficacy.

## 1. Introduction

Osteoarthritis (OA) is the most common cause of pain and disability worldwide, especially in the elderly population [1,2]. The hip and knee joints are most commonly affected, and women are more frequently affected than men [3,4]. Osteoarthritis is characterized by cartilage and meniscus degeneration, subchondral bone remodeling, synovial membrane inflammation, and infrapatellar fat pad inflammation and fibrosis [4,5].

There is no cure for OA, despite its high prevalence and deleterious impact on the quality of life of affected individuals [1,2]. The currently available therapeutic measures aim to relieve pain and maximize functional capacity and quality of life, while minimizing any adverse effects from drugs and invasive interventions [6]. Patients suffering from primary knee OA (pkOA) often report refractory, severe, and disabling pain and are eventually referred for partial or total arthroplasty [7].

Intra-articular (i.a.) injection of platelet-rich plasma (PRP), corticosteroids (CS), and hyaluronic acid (HA) is commonly used to manage pkOA [8,9]. These injections can easily be administered, fewer treatment sessions are necessary compared with other treatments, and the treatment adherence of patients is relatively easy to achieve (except for repeated injections). According to the Osteoarthritis Research Society International (OARSI) guidelines, i.a. injection of CS and HA (but not of PRP) are among the recommended treatments for pkOA, dependent upon comorbidity status [10]. A recent meta-analysis of the efficacy and safety of i.a. injection of CS and HA for pkOA concluded that both therapies are relatively safe and lead to comparable improvements in knee function [11]. According to such meta-analysis, better short-term effects (up to one month) are obtained with CS than with HA, and better long-term effects (up to six months) are achieved with HA than with CS [11]. On the other hand, i.a. injection of HA causes more topical adverse effects than i.a. injection of CS [11]. 

Over the last two decades, various studies (summarized in Appendix A; refs [12,13,14,15,16,17,18,19,20,21,22,23,24,25,26,27,28,29,30,31,32,33,34,35,36,37,38,39,40,41,42,43,44,45,46,47,48,49,50,51,52,53,54,55,56,57,58,59,60,61,62,63,64,65,66,67,68,69,70,71,72,73,74,75,76,77,78,79,80,81,82,83,84,85,86,87,88,89,90,91,92,93,94,95,96,97,98,99,100,101,102,103,104,105,106,107,108,109,110,111,112,113,114,115,116,117,118,119,120,121,122,123,124,125,126,127,128,129,130,131,132,133,134,135,136,137,138,139,140,141,142,143,144,145,146,147,148,149,150,151,152,153,154,155,156,157,158,159,160,161,162,163,164,165,166,167,168,169,170,171,172,173,174,175,176,177,178,179,180,181,182,183,184,185,186,187,188,189,190,191,192,193,194,195]) have proposed the management of pkOA with different types of stem cells, which can be derived from many different sources (autologous, adipose-derived regenerative cells; autologous, adipose-derived stem cells (obtained by culturing ADRCs); autologous stem cells isolated from infrapatellar fat pad; allogeneic, adipose-derived stem cells; autologous, micro-fragmented fat (from liposuction); autologous, centrifuged liposuction liquid; autologous bone marrow concentrate; autologous, bone-marrow-derived mesenchymal stem cells; allogeneic, bone-marrow-derived mesenchymal stromal cells; autologous, matrix-induced mesenchymal stem cells from synovia; allogeneic chondrocytes that overexpress transforming growth factor beta; allogeneic, human umbilical cord-derived MSCs; allogeneic, placental mesenchymal stem cells; autologous, activated peripheral blood stem cells; and allogeneic amniotic fluid). It is beyond the scope of this study to provide a comprehensive description of the different motivations behind the use of these different types of stem cells and their specific advantages and disadvantages. A general discussion of the advantages of autologous stem cells over allogeneic stem cells, as well as of the advantages of uncultured stem cells over cultured stem cells, is provided in, e.g., [196]. 

In general, there are various motivations behind the idea to manage pkOA with stem cells (summarized in, e.g., [197]). The most important ones are the findings that stem cells can be converted into chondrocytes in vitro, as well as that transplantation of stem cells prevented or improved experimental osteoarthritis in animal models (summarized in, e.g., [197]). On the other hand, it has remained unclear whether (and if so, to what extent) implanted stem cells transform to chondrocytes, replace missing cells, and thereby repair the cartilage (note that, in principle, this could only be achieved with autologous stem cells but not with allogeneic stem cells [196]). Alternatively, stem cells may suppress synovial activation and indirectly ameliorate cartilage damage by establishing a repair microenvironment and stimulating tissue repair through the recruitment of local endogenous stem cells [197]. 

As outlined in detail in Section 2 below, we identified 19 meta-analyses by means of a systematic literature search according to the PRISMA (Preferred Reporting Items for Systematic Reviews and Meta-Analyses) guidelines [198]. These papers were all published between January 2020 and July 2021 [199,200,201,202,203,204,205,206,207,208,209,210,211,212,213,214,215,216,217]. Unfortunately, the conclusions of these meta-analyes are inconsistent. Specifically, these meta-analyses demonstrated (i) the efficacy of treating pkOA with stem cells [199,200,201,202,203,204]; (ii) the superiority of autologous stem cells derived from adipose tissue over other types of stem cells in the management of pkOA [205,206,207,208,209]; (iii) the efficacy of treating pkOA with autologous, adipose-derived stem cells, without comparison with other types of stem cells [210,211]; (iv) the superiority of autologous, bone-marrow-derived stem cells over autologous, adipose-derived stem cells in the management of pkOA [212]; (v) the efficacy of treating pkOA with autologous, bone-marrow-derived stem cells, without comparison with other types of stem cells [213]; (vi) the efficacy of treating pkOA with allogeneic stem cells [214]; (vii) the efficacy of treating pkOA with stem cells only in conjunction with surgery [215]; and (viii) the lack of efficacy of treating pkOA with stem cells [216,217] (summarized in Appendix A).

To determine the reasons for these inconsistent conclusions, we performed a comprehensive assessment of all clinical studies included in these 19 meta-analyses [199,200,201,202,203,204,205,206,207,208,209,210,211,212,213,214,215,216,217]. Furthermore, we determined whether a separate meta-analysis based on an independent, systematic assessment of studies on the management of pkOA with stem cells would reach a different conclusion.

## 2. Materials and Methods

PubMed, Web of Science, and the Cochrane Library were searched for “knee osteoarthritis stem cell*”, “knee osteoarthritis stromal vascular fraction”, and “knee osteoarthritis SVF” from the days of inception of these databases until 7 August 2021 according to the PRISMA (Preferred Reporting Items for Systematic Reviews and Meta-Analyses) [198] guidelines. Duplicates were excluded. This search strategy found many more clinical studies than evaluated in the 19 assessed meta-analyses [199,200,201,202,203,204,205,206,207,208,209,210,211,212,213,214,215,216,217].

The strategy of the first assessment of the identified publications is summarized in Figure 1. 

In the first step, for each identified publication, it was determined by reading the title and abstract whether the publication represented a meta-analysis of clinical studies on the management of pkOA with stem cells and was published in 2020 or 2021. This was independently undertaken by C.S. and C.A. Results were compared and discussed until agreement was achieved. Afterwards, all studies included in the identified meta-analyses that reported the management of pkOA with stem cells were classified with regard to the type of study as summarized in Table 1 and Table 2, as well as with regard to the type of stem cells used as summarized in Table 3. Thereafter, all identified meta-analyses were classified by C.S. and C.A. with regard to the type of meta-analysis performed as summarized in Table 4.

Then, two types of meta-analysis were excluded from further assessment: those in which only the endpoints of the same patients before and after treatment were compared (four meta-analyses) (Class 2 in Table 4) and those in which the vast majority (>80%) of clinical studies included addressed management of pkOA with different modalities than stem cells (three meta-analyses) (Class 3 in Table 4).

Afterwards, the quality of each sub-analysis performed in the remaining 12 meta-analyses (Class 1 in Table 4) was assessed by C.S. and C.A. according to the quality criteria outlined in Table 5. 

The strategy of the second assessment of the identified publications is summarized in Figure 2. In the first step, C.S. and C.A. excluded reviews and investigations that did not represent clinical studies on the management of pkOA with stem cells. Then, each study identified in this search was classified with regard to the type of study as summarized in Table 1 and Table 2 and with regard to the type of stem cells used as summarized in Table 3. Furthermore, it was determined which analyses could be performed that fulfilled all quality criteria summarized in Table 5. All this was undertaken by C.S. and C.A.

## 3. Results

### 3.1. Assessment of Published Meta-Analyses

A total of 56 clinical studies [12,13,14,15,16,17,18,19,20,21,22,23,24,25,28,29,30,32,37,38,39,40,46,47,50,51,52,53,55,56,62,63,64,65,67,69,81,82,98,99,100,105,110,111,112,115,118,119,137,156,157,161,218,219,220,221] was included in the 19 assessed meta-analyses; details of these 56 clinical studies are provided in Appendix A. Classification of these 56 studies with regard to the type of study is summarized in Table 1 and Table 2. Two of these 56 studies [220,221] (3.6%) could not be evaluated because they were not listed in PubMed, Web of Science, and the Cochrane Library, nor are they listed on Google Scholar. 

Of note, four of the 54 studies [156,157,161,218] that could be evaluated (7.4%) did not address pkOA but rather, respectively, focal chondral, osteochondral, meniscal chondral, or meniscal lesions. Furthermore, only 29 of the 54 studies that could be evaluated (53.7%) were randomized controlled trials (RCT); the other studies were RCTs with the contralateral knee as internal control (3/54 = 5.6%), prospective cohort studies (7/54 = 13.0%), retrospective cohort studies (2/54 = 3.7%), and case series without control groups (13/54 = 24.1%) (Table 2).

Thirteen different types of stem cells (eight were autologous cell types, and five were allogeneic cell types) were applied in the 54 studies that could be evaluated (Table 3). The most frequently applied cell types were autologous, adipose-derived stem cells (ADSCs) (11/54 = 20.4%), autologous, bone-marrow-derived mesenchymal stromal cells (BM-MSCS) (10/54 = 18.5%), autologous, adipose-derived regenerative cells (ADRCs) (9/54 = 16.7%), and autologous bone marrow aspirate concentrate (BMAC) (7/54 = 13.0%). In one study [219], no cells but allogeneic amniotic fluid were applied.

Each of the 19 assessed meta-analyses included an average number of 9.3 ± 4.6 (mean ± standard deviation) studies (median, 8; range, 2–18). Conversely, each study listed in Appendix A was included in an average of 3.1 ± 3.5 of the 19 assessed meta-analyses (median, 2; range, 1–15). Only 9 of the 56 studies (16.0%) were included in five or more of the 19 assessed meta-analyses, whereas 26 of the 56 studies (46.4%) were included in only one of the 19 assessed meta-analyses. None of the 56 studies were included in all assessed meta-analyses; the most frequently included study, i.e., [20], was an RCT in which injection of allogeneic BM-MSCs was compared with injection of HA (considered in 15 of the 19 assessed meta-analyses). 

In four of the 19 assessed meta-analyses [202,207,210,212], subjective (self-reported pain and function scores) and objective (cartilage measurements) endpoints of the same patients before and after treatment were analyzed. Since this approach cannot rule out the possibility that all reported and measured effects were based on the placebo effect in the management of pkOA [222,223,224], these meta-analyses were excluded from further assessment.

In the three network meta-analyses, respectively, only two [217], three [204], and six [211] studies on the management of pkOA with stem cells were included, but, respectively, 41 [217], 22 [204], and 37 [211] studies addressing the management of pkOA without injection of stem cells were included. Since this approach does not at all reflect the literature that is actually available on the management of pkOA with stem cells (c.f. Appendix A), these network meta-analyses were also excluded from further assessment.

In the remaining 12 meta-analyses [199,200,201,203,205,206,208,209,213,214,215,216], a total of 157 sub-analyses were performed; details of these 157 sub-analyses and individual assessment according to the quality criteria listed in Table 5 are provided in Appendix A. The absolute and relative numbers of sub-analyses that fulfilled, respectively, 0/1/2/3/4/5/6/7/8 of the quality criteria outlined in Table 5 were 0/0/2/3/9/18/94/30/1 (or 0%/0%/1.3%/1.9%/5.7%/11.5%/59.9%/19.1%/0.6%, respectively). 

The two sub-analyses which fulfilled only 2 of the 8 quality criteria outlined in Table 5 are listed in Appendix A; they compared the following studies with respect to the VAS pain score at three months and six months post treatment: a study in which the management of pkOA was performed using allogeneic amniotic fluid [219], a RCT in which the application of allogeneic ADSCs was compared with placebo treatment [16] (study considered two times because of two different doses of stem cells), a RCT in which the application of autologous BM-MSCs plus HA injection was compared with HA injection alone [53] (this study was also considered two times because of two different doses of stem cells), and a study in which the application of autologous BMC plus platelet-poor plasma injections was compared with a placebo treatment, with the contralateral knee as internal control [62]. 

The only sub-analysis fulfilling all quality criteria outlined in Table 5 is listed in Appendix A, and compared the following studies with respect to categorical data (worse/not worse) of different MRI evaluations, which were categorical structures with different scales and, thus, converted to categorical data by the authors of this meta-analysis [213]: a RCT in which the management of pkOA with allogeneic placental mesenchymal stem cells was compared with placebo treatment [17], and a RCT in which the management of pkOA with allogeneic ADSCs was also compared with placebo treatment [16].

### 3.2. Systematic Assessment of Clinical Studies on Treatment of Primary Knee Osteoarthritis with Stem Cells

Our independent literature search yielded 2912 publications, of which 174 described clinical studies with a total number of *n* = 7146 patients treated with different types of stem cells (treatment groups). The other 2738 publications were duplicates (*n* = 858) or addressed topics other than the management of pkOA with stem cells (*n* = 681), reviews of the literature (*n* = 476), animal studies (*n* = 329), in vitro studies (*n* = 285), commentaries (*n* = 35), combined animal and in vitro studies (*n* = 26), conference abstracts (*n* = 17), study protocols (*n* = 11), animal studies on different topics (*n* = 6), reviews of different topics (*n* = 5), position statements (*n* = 4), errata (*n* = 3), and retraction notes (*n* = 2). Eight of the studies listed in Appendix A [12,15,22,25,50,55,112,156] were not found in this literature search, as well as one study [179] that was mentioned but not used in the sub-analyses performed in one of the assessed meta-analyses [199]. In these nine additional studies, a total number of *n* = 532 patients was treated with different types of stem cells (treatment groups). Accordingly, the combined literature search (all assessed meta-analyses and the literature search outlined in Figure 2) yielded a total of 183 studies, with a total number of *n* = 7678 patients. Details of these 183 studies are provided in Appendix A. Categorization of these studies with respect to treatment and control treatment is provided in Table 6, with respect to the type of study in Table 7 and with respect to the types of stem cells used in Table 8. Results of the corresponding combined three-step categorization are summarized in Table 9.

Most of the 183 studies listed in Appendix A were performed using autologous BMAC (38 studies comprising a total of 2905 patients, among them 2588 patients with pkOA), followed by autologous ADRCs (36 studies; 1608 patients; 1568 with pkOA), autologous BM-MSCs (34 studies; 468 patients; 385 patients with pkOA), autologous ADSCs (22 studies; 368 patients; 346 patients with pkOA), micro-fragmented fat from liposuction (MFF) (21 studies; 1489 patients with pkOA), peripheral blood-derived MSCs (8 studies; 183 patients with pkOA; 54 patients with pkOA), and other cell types (24 studies; 657 patients; 430 patients with pkOA). It is of note that in none of these 183 studies was a serious adverse event after application of stem cells reported.

Management of pkOA with stem cells was performed on a total of *n* = 6860 patients in 143 of the 183 studies (Categories I–V in Table 6). The remaining 40 clinical studies (including a total of 818 patients) addressed, respectively, focal chondral, osteochondral, meniscal chondral, or meniscal lesions (Category VI in Table 6), pathologies which are not addressed in the present setting. 

In 65 of these 143 clinical studies on the management of pkOA, treatment with stem cells was compared to sham treatment or another treatment (RCTs or prospective or retrospective two- or three-cohort studies, respectively) (Categories I–IV in Table 6). On the other hand, the management of pkOA with i.a. injection of stem cells as the sole treatment (not considering rehabilitation) was performed in only 38 of these 65 studies (Categories I-III in Table 6), and i.a. injection of, respectively, saline or sham treatment was only performed in 8 of these 38 studies (Category I in Table 6). 

According to the quality criteria of the meta-analyses of studies on the management of pkOA with stem cells outlined in Table 5, it would generally be possible to perform the sub-analyses shown in Table 10. However, MFF must not be confused with ADRCs [225,226], and there are substantial differences between cultured stem cells derived from various human tissues [227]. The latter argument also applies to the 56 studies included in the 19 assessed meta-analyses, which further diminishes the value of these meta-analyses as an assessment of the management of pkOA with stem cells. 

Furthermore, a meta-analysis requires that, in all included studies, the same type of data is reported (i.e., same assessment scores; same or very similar intervals between treatment and follow-up; and mean, standard deviation, and number of patients in each group). As shown in Table 11, this was not achieved for the clinical studies on the management of pkOA with autologous ADRCs, autologous ADSCs, and autologous BM-MSCs listed in Table 10.

## 4. Discussion

Concerning the first quality criterion outlined in Table 5, it should be mentioned that a meta-analysis is a statistical analysis that combines the results of multiple scientific studies. Hence, calculating the overall effects based on a single study should not be considered a meta-analysis. However, calculating the overall effects based on a single study is exactly what was performed in 5 of the 19 assessed meta-analyses [206,209,214,215,216] and 16 of the 157 (10.2%) sub-analyses listed in Appendix A.

With respect to the second and third quality criteria outlined in Table 5, it appears obvious that, in meta-analyses of studies in which the management of pkOA with stem cells was investigated, only studies focusing on the management of pkOA and applying stem cells should be included. However, this was not always the case in the 19 assessed meta-analyses. The pathomechanisms of focal chondral, osteochondral, or meniscal chondral lesions [156,157,161], as well as of injuries that require partial medial meniscectomy [86], are not the same as the pathomechanisms of pkOA. This will likely impact on treatment outcomes when applying stem cells. Furthermore, despite the fact that stem cells can be isolated from amniotic fluid [228], in our opinion, the injection of allogeneic amniotic fluid itself [219] should not be considered a stem cell treatment, as it does not contain, by definition, stems cells, though it may contain products of secretion by stem cells.

Concerning the fourth quality criterion outlined in Table 5, non-randomized clinical trials and studies in which the contralateral knee was used as an internal control may differ from RCTs in terms of selection bias (as a result of systematic differences between the baseline characteristics of the groups that are compared [229]), performance bias (as a result of systematic differences between groups in the care that is provided or in exposure to factors other than the interventions of interest [230]), and attrition bias (as a result of systematic differences between groups in withdrawals from a study [229]). Accordingly, non-randomized clinical trials and studies in which the contralateral knee was used as an internal control should not be combined with RCTs in meta-analyses. However, this quality criterion was not fulfilled in 24 of the 157 (15.3%) sub-analyses performed in the twelve assessed meta-analyses (Category 1 in Table 4). 

With respect to the fifth quality criterion outlined in Table 5, it is crucial to bear in mind that studies in which the management of pkOA with stem cells was compared to placebo treatment (or studies in which the management of pkOA with stem cells plus a concomitant therapy was compared to the concomitant therapy alone, respectively) must not be combined with studies in which the management of pkOA with stem cells (with or without concomitant therapy) was compared to a different treatment. However, this quality criterion was not fulfilled in 102 of the 157 (65.0%) sub-analyses performed in the twelve assessed meta-analyses. Specifically, in 86 of the 157 (54.8%) sub-analyses performed in the twelve assessed meta-analyses, studies in which the management of pkOA with stem cells was compared with placebo treatment (or studies in which the management of pkOA with stem cells plus concomitant therapy was compared with the concomitant therapy alone, respectively) were combined with studies in which the management of pkOA with stem cells was compared with the injection of HA. However, this approach disregards the documented, positive effects of injection of HA as the management of pkOA [11] and may thus substantially underestimate the positive effects of treating pkOA with stem cells in meta-analyses. Furthermore, in 9 of the 157 (5.7%) sub-analyses performed in the twelve assessed meta-analyses, only studies in which the management of pkOA with stem cells was compared with the management of pkOA with the injection of HA were included. Due to the documented, positive effects of the injection of HA injection as the management of pkOA [11], these sub-analyses may reach a different conclusion than sub-analyses in which only studies in which the management of pkOA with stem cells was compared with placebo treatment were included (or studies in which the management of pkOA with stem cells plus concomitant therapy was compared with the concomitant therapy alone, respectively).

Concerning the sixth quality criterion outlined in Table 5, it is of note that, in contrast to autologous cells, the application of allogeneic cells bears the risk of a HLA mismatch; compromised clinical outcomes after application of allogeneic cells were repeatedly reported in the literature [231,232,233]. In a position statement recently published by representatives of the U.S. Food and Drug Administration (FDA) in The New England Journal of Medicine [234], it was stated that autologous stem cells may typically raise fewer safety concerns than allogeneic stem cells. Consequently, studies using autologous stem cells should not be combined with studies using allogeneic stem cells in meta-analyses. However, this quality criterion was not fulfilled in 93 of the 157 (59.2%) sub-analyses performed in the twelve assessed meta-analyses. 

With respect to the seventh quality criterion outlined in Table 5, ADSCs and BM-MSCs are expanded in culture before application, whereas ADRCs and BMAC are not. As a consequence, the percentage of certain cell types in ADSCs fundamentally differs from the percentage of the same cell types in ADRCs [226], and the percentage of certain cell types in BM-MSCs fundamentally differs from the percentage of the same cell types in BMAC [235]. Hence, in meta-analyses of studies in which pkOA was treated with stem cells studies in which cultured cells were applied should not be combined with studies in which uncultured cells were applied. However, this quality criterion was not fulfilled in 24 of the 157 (15.3%) sub-analyses performed in the twelve assessed meta-analyses. 

Concerning the eighth quality criterion outlined in Table 5, it should be realized that each study has its own inclusion and exclusion criteria which may vary among studies. Furthermore, there are many different ways to prepare a certain type of stem cell (e.g., approximately 40 different approaches to isolate ADRCs from adipose tissue are reported in the literature, with substantial inter-individual variations in the final cell suspensions [225]). Accordingly, considering the same study more than once in a meta-analysis can introduce bias. For example, Maheshwer et al. [215] performed a sub-analysis of changes in mean cartilage volume after treatment of pkOA with adipose-derived stem cells in their meta-analysis (Appendix A). Four out of the five datasets used in this sub-analysis were taken from the same clinical study by Lu et al. [23]. Actually, these four datasets represented the same patients in four different ways (femoral cartilage volume of the left leg, femoral cartilage volume of the right leg, total cartilage volume of the left leg, and total cartilage volume of the right leg) (Figure 5 in the meta-analysis by Mashewer et al. [215]). Accordingly, this sub-analysis in the meta-analysis by Maheshwer et al. [215] could have come to a different conclusion if the study by Lu et al. [23] were considered only once in this sub-analysis. The eighth quality criterion outlined in Table 5 was not fulfilled in 57 of the 157 (36.3%) sub-analyses performed in the twelve assessed meta-analyses.

We cannot offer any explanation as to why the 19 assessed meta-analyses did not adhere to the quality criteria summarized in Table 5. As outlined in detail above, these quality criteria reflect general considerations about meta-analyses and the comparability of different types of stem cells with regard to their safety and regenerative potential.

For the sake of completeness, it should be mentioned that, recently, the application of stem cells isolated from infrapatellar fat pads was proposed for the management of pkOA, mostly due to their ability for differentiation into chondrocytes and constitutive immunomodulatory properties (reviewed in [236,237]). On the other hand, some authors have argued (based on in vitro investigations of stem cells isolated from the infrapatellar fat pad of patients suffering from pkOA) that these cells may not be effective to counteract NAD+-mediated inflammation, which is an important signature of osteoarthritis [238]. Another potential disadvantage of the application of stem cells isolated from infrapatellar fat pads is the need for surgery to harvest parts of the infrapatellar fat pad, which may add additional safety issues to the management of pkOA (potential contamination of the knee joint). Hence, clinical studies are necessary to demonstrate the potential advantages of stem cells isolated from human infrapatellar fat pads over other types of stem cells in the management of pkOA. In this regard, a very recent, first-in-human case series on 12 patients without a control group indicated that the management of pkOA with these cells is safe and effective [195]. However, the data presented in this study [195] do not indicate that stem cells isolated from human infrapatellar fat pads will outperform other types of stem cells in the management of pkOA. This can be seen from data provided in Table S2 in Supplementary Material of [195]. In this table, the authors provided individual Western Ontario and McMaster Universities Osteoarthritis Index (WOMAC) scores of the 12 patients at baseline and the improvement 24 weeks post-treatment. In order to compare their results with the results of other clinical studies mentioned in Appendix A in [195], the authors applied linear regression analysis (data from baseline and 24 weeks post-treatment) in order to calculate slopes of improvement and found that their approach resulted in the steepest regression line (and, thus, best outcome) compared to the other studies listed in Appendix A of [195]. However, it should be taken into account that, among those studies listed in Appendix A in [195], their own study reported the lowest mean WOMAC score at baseline (28.6) (higher WOMAC scores indicate worse pain, stiffness, and functional limitations). The other studies listed in Appendix A in [195] reported mean WOMAC scores at baselines of, respectively, 31.2 [29], 38.8 [38], and 69.0 [37]. Accordingly, one cannot exclude that the finding of the steepest regression line between the WOMAC scores at baseline and the WOMAC scores at 24 weeks post-treatment in [195] compared with the other studies listed in Appendix A in [195] was influenced by the fact that the patients in [195] had the best starting conditions for experiencing treatment success. Furthermore, the authors of [195] could have compared their data with results reported in [19] (not cited in [238]). In [19], a decrease in the mean WOMAC score from 50 at baseline to 10 at 24 weeks post treatment was reported, resulting in a slope of improvement that outperformed the slope of improvement reported in [195]. The key difference between [19] and [195] is the fact that patients were treated with uncultured ADRCs in [19] but with cultured stem cells isolated from infrapatellar fat pads in [195]. The specific advantages of uncultured ADRCs over cultured stem cells for regeneration in musculoskeletal disorders are outlined in detail in [196]. In summary, the current literature does not indicate any specific advantages of stem cells isolated from human infrapatellar fat pads over other types of stem cells in the management of pkOA. 

This review had two limitations. First, only PubMed, Web of Science, and the Cochrane Library were searched. However, considering the sophisticated search strategies described in the 19 assessed meta-analyses and inclusion of all studies that were included in the 19 assessed meta-analyses also in the present investigation minimized the risk to overlook any relevant clinical study on the management of pkOA with stem cells. Second, no meta-analyses published before 2020 were assessed. However, this does not devalue our finding that the 19 meta-analyses published from January 2020 to July 2021 were not scientifically sound.

## 5. Conclusions

The inconsistent conclusions of the 19 assessed meta-analyses regarding the efficacy and safety of treating pkOA with stem cells were most probably based on substantial inter-individual differences in literature search strategies among different authors, misconceptions about meta-analyses themselves, and misconceptions about the comparability of different types of stem cells with regard to their safety and regenerative potential. None of the 19 assessed meta-analyses should be considered as being able to provide a scientifically definitive assessment of the efficacy of treating pkOA with stem cells. Accordingly, clinicians should be cautious of the 19 assessed meta-analyses of clinical studies on the management of pkOA with stem cells and the conclusions drawn therein and strive to participate in FDA and/or EMA approved trials that have been reviewed to provide clinically and statistically valid efficacy.

Furthermore, at this time, it appears impossible to perform a scientifically sound meta-analysis of such a heterogenous set of studies which investigated the management of pkOA with stem cells. There are strong indications that the management of pkOA with stem cells is safe and effective (e.g., all RCTs in Category I in Appendix A reported the superiority of the management of pkOA over treatment with a placebo). However, apparently it is still too early for scientifically sound meta-analyses on this topic.

## Figures and Tables

**Figure 1 cells-11-00965-f001:**
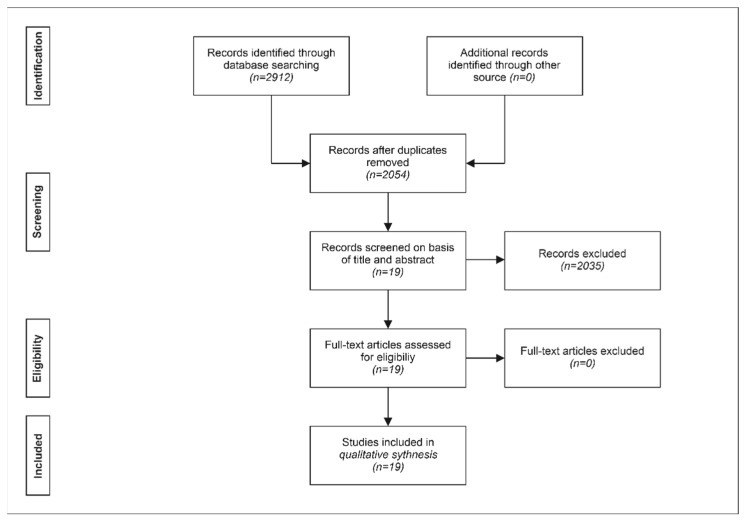
Systematic review flow chart of the first literature search regarding meta-analyses of clinical studies on the treatment of primary knee osteoarthritis with stem cells that were published between January 2020 and July 2021, performed according to the PRISMA guidelines [198] on 7 August 2021.

**Figure 2 cells-11-00965-f002:**
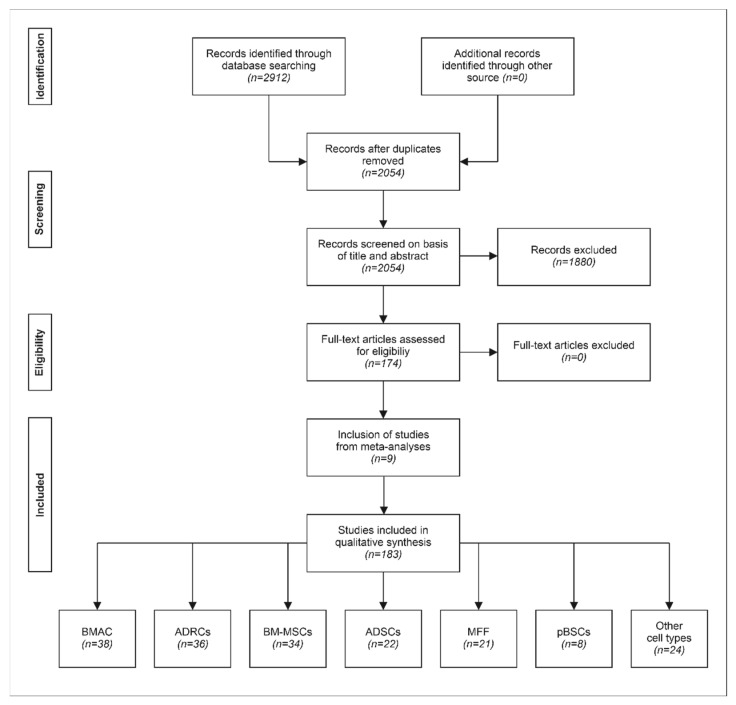
Systematic review flow chart of the second literature search regarding treatment of primary knee osteoarthritis with stem cells, performed according to the PRISMA guidelines [198] on 7 August 2021. Abbreviations: BMAC, bone marrow concentrate; ADRCs, adipose-derived regenerative cells; BM-MSCs, bone-marrow-derived mesenchymal stem cells; ADSCs, adipose-derived stem cells; MFF, micro-fragmented fat (from liposuction); and pBSCs, activated peripheral blood stem cells.

**Table 1 cells-11-00965-t001:** Categories of studies included in the 19 meta-analyses summarized in Appendix A, which details clinical studies in which management of primary knee osteoarthritis with stem cells was investigated.

C	N_all_	Description
I	8	Intra-articular injection of stem cells as the sole treatment (not considering rehabilitation) compared with i.a. injection of saline or sham treatment as control.
II	6	Intra-articular injection of stem cells as the sole treatment (not considering rehabilitation) compared with i.a. injection of, respectively, PRP, CS or HA as control.
III	10	Intra-articular injection of stem cells as the sole treatment (not considering rehabilitation) compared with other treatments than those in Categories I and II as control.
IV	12	Combinations of i.a. injection of stem cells with other modalities compared with sham treatment or other treatments as control.
V	13	Combinations of i.a. injection of stem cells with or without other modalities without control group (case series or case reports).
VI	4	Treatment of focal chondral, osteochondral, or meniscal chondral lesions with stem cells as the sole treatment (not considering rehabilitation) or combinations of stem cells and other modalities with or without other treatments as control.
VII	1	Treatments that did not comprise stem cells.
VIII	2	Study not listed in PubMed, Embase, Web of Science, Cochrane Library, or Google Scholar.
Sum	56	

Abbreviations: C, category; N_all_, number of studies among all studies included in the 19 assessed meta-analyses; pkOA, primary knee osteoarthritis; i.a., intra-articular; PRP, platelet-rich plasma; CS, corticosteroid; and HA, hyaluronic acid.

**Table 2 cells-11-00965-t002:** Types of studies included in the meta-analyses summarized in Appendix A, which details clinical studies in which management of primary knee osteoarthritis with stem cells was investigated.

Type	N_all_	N_pkOA_	Description
a	29	25	Randomized controlled trials (RCTs)
b	3	3	RCTs with the contralateral knee as internal control
c	7	7	Prospective cohort studies
d	2	2	Retrospective cohort studies
e	13	13	Case series with more than one subject
f	0	0	Case reports with only one subject
N	2	2	Study not listed in PubMed, Embase, Web of Science, Cochrane Library, or Google Scholar
Sum	56	52	

Abbreviations: N_all_, number of studies among all studies included in the 19 assessed meta-analyses and N_pkOA_, number of studies among those studies included in the 19 assessed meta-analyses that addressed pkOA.

**Table 3 cells-11-00965-t003:** Cell types used in those studies included in the meta-analyses summarized in Appendix A in which treatment of primary knee osteoarthritis with stem cells was investigated.

Cell Type	O	N_all_	N_pkOA_	Description
ADRCs	Auto	9	9	Autologous, adipose-derived regenerative cells
ADSCs	Auto	11	11	Autologous, adipose-derived stem cells (obtained by culturing ADRCs)
ADSCs	Allo	1	1	Allogeneic, adipose-derived stem cells
MFF	Auto	4	4	Autologous, micro-fragmented fat (from liposuction)
CLL	Auto	1	1	Autologous, centrifuged liposuction liquid
BMAC	Auto	7	7	Autologous bone marrow concentrate
BM-MSCs	Auto	10	9	Autologous, bone-marrow-derived mesenchymal stem cells
BM-MSCs	Allo	3	2	Allogeneic, bone-marrow-derived mesenchymal stromal cells
S-MSCs	Auto	1	0	Autologous, matrix-induced MSCs from synovia
Ch-TGFβ	Allo	2	2	Allogeneic chondrocytes that overexpress transforming growth factor beta
hUC-MSCs	Allo	2	2	Allogeneic, human-umbilical-cord-derived MSCs
P-MSCs	Allo	1	1	Allogeneic, placental MSCs
pBMCs	Auto	1	0	Autologous, activated peripheral blood stem cells
No cells	Allo	1	1	Allogeneic amniotic fluid
Sum		54	50	

Abbreviations: N_all_, number of studies among all studies included in the 19 assessed meta-analyses and N_pkOA_, number of studies among those studies included in the 19 assessed meta-analyses that addressed pkOA.

**Table 4 cells-11-00965-t004:** Type of analysis performed in the meta-analyses summarized in Appendix A in which treatment of primary knee osteoarthritis with stem cells was investigated.

C	N	Description
1	12	Meta-analysis of studies in which treatment of pkOA with stem cells was compared with placebo treatment (or studies in which treatment of pkOA with stem cells plus concomitant therapy was compared with the concomitant therapy alone, respectively).
2	4	Meta-analysis in which only endpoints of the same patients before and after treatment were compared.
3	3	Network meta-analysis that included only a small number of studies on the treatment of pkOA with stem cells and a much higher number of studies on the treatment of pkOA without stem cells.
Sum	19	

Abbreviations: C, category; N, number of meta-analyses; i.a., intra-articular; pkOA, primary knee osteoarthritis; PRP, platelet rich plasma; CS, corticosteroid; and HA, hyaluronic acid.

**Table 5 cells-11-00965-t005:** Criteria used to assess the quality of sub-analyses in the meta-analyses summarized in Appendix A of studies in which treatment of primary knee osteoarthritis with different types of stem cells was investigated.

No.		N_a_	N_r_ [%]
1	At least 2 different clinical studies were included.	141	89.8
2	Only clinical studies on pkOA were included.	141	89.8
3	Only clinical studies in which stem cells were applied were included.	155	98.7
4	Only randomized controlled trials were included.	133	84.7
5	Only clinical studies were included in which application of stem cells was compared with placebo treatment or in which application of stem cells plus concomitant therapy (including arthroscopic debridement, high tibial osteotomy, injection of hyaluronic acid, etc.) was compared with the concomitant therapy alone, respectively.	55	35.0
6	Only clinical studies using, respectively, autologous or allogeneic stem cells were included.	64	40.8
7	Only clinical studies in which, respectively, cultured or uncultured cells were applied were included.	133	84.7
8	Clinical studies in which more than 1 dose of stem cells was applied were only considered once in the corresponding meta-analysis.	100	63.7

Abbreviations: N_a_ and N_r_, absolute (NA) and relative (NR) number of sub-analyses that fulfilled the corresponding criterion; No., number of criteria; and pkOA, primary knee osteoarthritis.

**Table 6 cells-11-00965-t006:** Categories of studies on treatment of primary knee osteoarthritis with stem cells that were identified during an evidence-based, systematic review of the literature according to the PRISMA guidelines [198] performed on 7 August 2021 (summarized in Appendix A).

C	N_all_	Description
I	8	Treatment of pkOA with i.a. injection of stem cells as the sole treatment (not considering rehabilitation) compared with i.a. injection of saline or sham treatment as control.
II	8	Treatment of pkOA with i.a. injection of stem cells as the sole treatment (not considering rehabilitation) compared with i.a. injection of, respectively, PRP, CS or HA as control.
III	22	Treatment of pkOA with i.a. injection of stem cells as the sole treatment (not considering rehabilitation) compared with other treatments than those in Categories I and II as control.
IV	27	Treatment of pkOA with combinations of stem cells and other modalities compared with sham treatment or other treatments as control.
V	78	Treatment of pkOA with combinations of stem cells with or without other modalities without control group (case series or case reports).
VI	40	Treatment of focal chondral, osteochondral, or meniscal chondral lesions with stem cells as the sole treatment (not considering rehabilitation) or combinations of stem cells and other modalities with or without other treatments as control.
Sum	183	

Abbreviations: C, category; N, number of studies; i.a., intra-articular; pkOA, primary knee osteoarthritis; PRP, platelet rich plasma; CS, corticosteroid; and HA, hyaluronic acid.

**Table 7 cells-11-00965-t007:** Types of studies on treatment of primary knee osteoarthritis with stem cells that were identified during an evidence-based, systematic review of the literature according to the PRISMA guidelines [198] performed on 7 August 2021.

Type	N_all_	N_pkOA_	Description
a	44	33	Randomized controlled trials (RCTs)
b	6	6	RCTs with the contralateral knee as internal control
c	18	15	Prospective cohort studies
d	12	11	Retrospective cohort studies
e	92	73	Case series with more than one subject
f	11	5	Case reports with only one subject
Sum	183	143	

Abbreviations: N_all_, number of studies among all studies listed in Appendix A, and N_pkOA_, number of studies among those studies listed in Appendix A that addressed primary knee osteoarthritis (pkOA).

**Table 8 cells-11-00965-t008:** Types of stem cells used in studies on management of primary knee osteoarthritis with stem cells that were identified during an evidence-based, systematic review of the literature according to the PRISMA guidelines [198] performed on 7 August 2021.

Cell Type	N_all_	N_pkOA_	Description
ADRCs	36	35	Adipose-derived regenerative cells
ADSCs	22	18	Adipose-derived stem cells (obtained by culturing ADRCs)
MFF	21	21	Micro-fragmented fat (from liposuction)
CLL	1	1	Centrifuged liposuction liquid
BMA	1	1	Bone marrow aspirate
BMAC	38	26	Bone marrow aspirate concentrate
BM-MSCs	34	25	Bone-marrow-derived mesenchymal stem cells
Cs/CPs	2	0	Chondrocytes and chondrocyte precursors
CSCs	1	0	Cartilage stem cells
MACI	1	0	Matrix-induced, autologous chondrocyte implant
S-MSCs	4	1	Matrix-induced MSCs from synovia
Ch-TGFβ	2	2	Chondrocytes that overexpress transforming growth factor beta
hUC-MSCs	4	3	Human-umbilical-cord-derived MSCs
hUCB-MSCs	7	6	Human-umbilical-cord-blood-derived MSCs
P-MSCs	1	1	Placental MSCs
pBSCs	8	4	Activated peripheral blood stem cells
Sum	183	143	

Abbreviations: N_all_, number of studies among all studies listed in Appendix A, and N_pkOA_, number of studies among those studies listed in Appendix A that addressed primary knee osteoarthritis (pkOA).

**Table 9 cells-11-00965-t009:** Detailed analysis of clinical studies on the management of primary knee osteoarthritis with stem cells (summarized in Appendix A) that were identified during an evidence-based, systematic review of the literature according to the PRISMA guidelines [198] performed on 7 August 2021. The categories of studies (I to VI) are explained in Table 6 and the types of study (a to f) in Table 7.

Category	Type of Study	ADRCs	ADSCs	MFF	CLL	BMA	BMAC	BM-MSCs	Cs/CPs	CSCs	MACI	S-MSCs	Ch-TGFβ	hUC-MSCs	hUCB-MSCs	P-MSCs	pBSCs	Sum
I	a	1	2					1					2			1		7
	b						1											1
	c																	0
	d																	0
II	a		1	1			2	1						2				7
	b																	0
	c																	0
	d	1																1
III	a		4				1	1										6
	b						3											3
	c	1	5	2				1										9
	d	2		1				1										4
IV	a	2	1	1			1	7									1	13
	b	1					1											2
	c	5	1															6
	d	1		1			4											6
V	e	20	3	15	1	1	13	11				1		1	5		2	74
	f	1	1					2										4
VI	a	1	1				1	3				1			1		3	11
	b																	0
	c						3											3
	d						1											1
	e		1				6	4	1	1	1	2		1	1	1		19
	f		2				1	2	1									6
N_pkOA_		1568	346	1489	20	3	2588	385				8	128	75	186	10	54	6860
N_CL_		40	22				317	83	13	15	15	18		*	166		129	818

General abbreviations: N_pkOA_, number of patients with primary knee osteoarthritis treated in these studies; N_CL_, number of patients with chondral lesions treated in these studies; and *, number of patients not provided. Abbreviations of cell types: ADRCs, adipose-derived regenerative cells; ADSCs, adipose-derived stem cells (obtained by culturing ADRCs); MFF, micro-fragmented fat (from liposuction); CLL; centrifuged liposuction liquid; BMA, bone marrow aspirate; BMAC, bone marrow aspirate concentrate; BM-MSCs, bone-marrow-derived mesenchymal stromal cells; Cs, chondrocytes; CPs, chondrocyte precursors; CSCs, cartilage stem cells; MACI; matrix-induced autologous chondrocyte implant; S-MSCs, matrix-induced MSCs from synovia; hUC-MSCs, human umbilical cord-derived MSCs; hUCB-MSCs, human umbilical cord blood-derived MSCs; P-MSCs, placental MSCs; and pBSCs, activated peripheral blood stem cells.

**Table 10 cells-11-00965-t010:** Sub-analyses that would generally be possible in a meta-analysis of studies on treatment of primary knee osteoarthritis with stem cells (summarized in Appendix A) that were identified during an evidence-based, systematic review of the literature according to the PRISMA guidelines [198] performed on 7 August 2021.

R	C	T	First Author	Y	Cell Type	Treatment	Control
**Autologous, uncultured cells**							
[19]	1	a	Garza	2020	ADRCs	C	RS
[52]	4	a	Koh	2014	ADRCs	C + AD + HTO + PRP	AD + HTO + PRP
[58]	4	a	Peretti	2018	MFF	C + AD	AD
**Autologous, cultured cells**							
[18]	1	a	Lee	2019	ADSCs	C	Sa
[30]	3	a	Freitag	2019	ADSCs	C	CM
[56]	4	a	Zhang	2018	ADSCs	C + HA	C
[60]	4	a	Qiao	2020	ADSCs	C + AD + MF + HA	MF or MF + HA
[14]	1	a	Emamedin	2018	BM-MSCs	C	Sa
[50]	4	a	Varma	2010	BM-MSCs	C + AD	AD
[51]	4	a	Wong	2013	BM-MSCs	C + MF + HTO + HA	MF + HA + HTO
[53]	4	a	Lamo-Espinosa	2016	BM-MSCs	C + HA	HA
[55]	4	a	Bastos	2018	BM-MSCs	C + PRP	C
[57]	4	a	Lamo-Espinosa	2018	BM-MSCs	C + HA	HA
[59]	4	a	Lamo-Espinosa	2020	BM-MSCs	C + PRP	PRP
**Allogeneic, cultured cells**							
[12]	1	a	Cherian	2015	Ch-TGFß	C	Sa
[13]	4	a	Gupta	2016	BM-MSCs	C + HA	Sa + HA
[15]	1	a	Kim	2018	Ch-TGFß	C	Sa
[16]	1	a	Kuah	2018	ADSCs	C	Sa
[17]	1	a	Khalifeh Soltani	2019	P-MSCs	C	Sa

General abbreviations: R, reference number; C, category of study as outlined in Table 6; T, type of study as outlined in Table 6; and Y, year of publication. Abbreviations of cell types (in alphabetical order): ADRCs, adipose-derived regenerative cells; ADSCs, adipose-derived stem cells; BM-MSCs, bone-marrow-derived mesenchymal stromal cells; Ch-TGFß, chondrocytes that overexpress transcription growth factor beta; MFF, micro-fragmented fat; P-MSCs, placental MSCs. Abbreviations of treatments (in alphabetical order): AD, arthroscopic debridement; C, cells; HTO, high tibial osteotomy; MF, microfracture; RS, Ringer solution; PRP, platelet rich plasma; and Sa, saline.

**Table 11 cells-11-00965-t011:** Reported outcome in studies on treatment of primary knee osteoarthritis with autologous, adipose-derived regenerative cells, or autologous, adipose-derived stem cells listed in Table 10.

R	FA	Reported Outcome
**Autologous ADRCs**		
[19]	Garza	WOMAC total score at BL and at W6, M3, M6 and M12 (mean, median, interquartile range, median percentage range, minimum, maximum) Cartilage loss (mean at BL, mean change at M6) Outerbridge classification (median at BL, range at BL, median change at M6)
[52]	Koh	VAS pain score (mean and SD at baseline and at last follow-up) KOOS sub-scores (mean improvement from BL to LFU) Lysholm score (mean and SD at BL and LFU) Weight-bearing line (%) (mean and SD at BL and LFU) Femorotibial angle (°) (mean and SD at BL and LFU)
**Autologous ADSCs**		
[18]	Lee	VAS pain score at BL, M3 and M6 (mean) WOMAC total and sub-scores at BL, M3 and M6 (mean) KOOS sub-scores at BL, M3 and M6 (mean) Size of cartilage defect in MRI at BL and M6 (mean, SD)
[30]	Freitag	VAS pain score at BL, M1,5, M3, M6 and M12 (mean, 95% CI) KOOS sub-scores at BL, M1,5, M3, M6 and M12 (mean, 95% CI) WOMAC score at BL, M1,5, M3, M6 and M12 (mean, 95% CI)
[56]	Zhang	VAS pain score at BL, M3, M6, M12, M24 and M36 (mean, SD) WOMAC total score at BL, M3, M6, M12, M24 and M36 (mean, SD)
[60]	Qiao	SF-36 Physical Component and Mental Component sub-scores at BL, M3, M6, M9, M12 and M24 (mean, SD)

Abbreviations: ADRCs, adipose-derived regenerative cells; ADSCs, adipose-derived stem cells; R, reference; FA, first author; BL, baseline; W6, 6 weeks post treatment; M3/M6/M9/M12/M24, three/six/nine/twelve/24 months post-treatment; LFU, last follow-up (range, M14–M24; mean: M19.8); SD, standard deviation; and CI, confidence interval.

## Data Availability

All relevant data are reported in the manuscript and Appendix A.

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
