# Peer review of "Methodological Flaws in Meta-Analyses of Clinical Studies on the Management of Knee Osteoarthritis with Stem Cells: A Systematic Review"

_cells, 2022, doi:10.3390/cells11060965_

Round 1
Reviewer 1 Report
In the manuscript entitled “Methodological flaws in meta-analyses of clinical studies on the management of knee osteoarthritis with stem cells: a systematic review” the authors evaluated 19 meta-analyses regarding the treatment of knee osteoarthritis with stem cells published between January 2020 and July 2021 which had inconsistent conclusions. In order to explain the reason for these inconsistencies, the authors made a great effort to identify the differences in methodology of these different studies. According to this meta-analysis, the varying results of the evaluated analyses were based on differences in literature search strategies among different authors.
This study is of great interest for clinicians treating osteoarthritis. Before publication, some minor concerns have to be addressed.
At the end of line 66, the number “13” is not correctly formatted as a literature reference. Similar for line 174.
In table 4 and table 9, in the description for “Ch-TGFβ”, I assume it should read “transforming growth factor beta" and not “transcription growth factor beta"?
In table 5, it should read “Nall”, not “N”
In table 6, it seems that the “relative (NR) number of sub-analyses that fulfilled the corresponding criterion” are referring to the same entirety of sub-analyses. In that case, the criteria No. 1 and No. 2 with the same absolute number cannot have different relative numbers. Additionally, if the entirety of sub-analyses is 157 (and it seems to be that way), then the relative number for criterion No. 8 must be 63.7 and not 63.4.
Reviewer 2 Report
The paper by Schmitz et al nicely analyses the methodological flaws in 19 recently published meta-analyses regarding stem cells use in knee osteoarthrisis. The paper is interesting, clear and sound. The authors clearly reported the limits of meta-analyses based on too heterogenous studies.
I suggest only minor revisions of few errors:
Page 2 ,line 66. Please correct “[…] function.13 […]”.
Page 6, Table 2 and 3. Please correct “[…] Googe Scholar […]”.
Page 9, Figure 2. Please correct “frrom”.
Page 9, line 173. Please correct “Figure 1”.
Page 9, line 174. Please correct “[…] guidelines14 […]”.
Page 15, Table 11. Please correct “S + HA”.
Page 15, line 352. Please correct “[…] S, reference number […]”.
Page 15, line 353 and line 355. Please correct “Sup-plemental” and “adi-pose”.
Page 17, line 427. A “)” is lacking.
Reviewer 3 Report
The reading of the paper is difficult. Try to summarize and to avoid repetitions where possible.
Introduction
The introduction has to be revised to better explain the pathogenesis of osteoarthritis disease contextualizing the contribution of all tissues. Indeed, osteoarthritis is characterized by cartilage and meniscus degeneration, subchondral bone remodeling, synovial membrane inflammation and infrapatellar fat pad inflammation and fibrosis (doi.org/10.1016/j.reth.2020.07.007; doi: 10.3390/ijms21176016).
Many therapies have been developed to treat osteoarthritis and cell therapy seems one of the most promising. Please add a paragraph describing the role of stem cells as a regenerative approach to treat osteoarthritis. Furthermore, explain the different sources of stem cells that can be isolated from several tissues including adipose tissue, bone marrow etc.
Line 66: “13” should be “[13]”.
Lines 72-75: move this part in materials and methods.
Materials and methods
Please better describe how the search has been done. Why mesenchymal and stromal cells but only knee osteoarthritis stem cell e knee osteoarthritis svf?
Page 9: please correct the number of the figure- this is figure 2 and not 1.
Discussion
Please add a paragraph regarding a new source of stem cells such as adipose stem cells derived from the infrapatellar fat pad (doi: 10.3389/fcell.2019.00323).
Lines 428-432: this part is unclear.
Tables
Tables need to be reformatted in order to provide a better visualization of the data reported.
Tables are difficult to read and sometimes contain several repetitions. Please try to summarize the data in order to facilitate the reader.
e.g.
Tab 1: In the heading authors reported “published between January 2020 and July 2021 “. Please delete as this information is already reported in the text. Similarly, please move “original quotes taken from the abstracts of the cited papers” in the heading without repetitions in all the subtitles
Furthermore, please summarize the conclusion of the studies not reporting the entire text.
Tab 2 describe only the treatment e.g injection of stem cells …. Please do not repeat “Treatment of pkOA”
Please place the abbreviations at the footnote of the tables and not in the heading.
TABLE 11 e 12
Other comments
In the abstract and in the conclusions, the authors wrote that there is a misconception about the biology of stem cells. This point needs to be clarified.
Supplemental Table 6 (cont.)
Please check this table number. It seems the continuation of table S1.
Round 2
Reviewer 3 Report
I am partially satified by the reply of the authors. I have still two major concerns.
In my previous report I asked to add a brief paragraph describing the role of stem cells as a regenerative approach to treat osteoarthritis and to briefly describe the different sources of stem cells that can be isolated from several tissues. However, the authors replied that they would add a paragraph about different OA therapies. This is not what I asked. As the journal is not a rheumatology/orthopedics journal, it is necessary to provide an adequate background so that the paper can be readable and understandable to readers of different disciplines.
Regarding the discussion, I am aware that the current literature does not still indicate specific advantages of stem cells isolated from human infrapatellar fat pad over other types of stem cells in the management of osteoarthritis. The purpose of a review is also to provide insights for future studies and show criticalities/weaknesses on the subject matter. For this reason, I suggested to briefly add/discuss this point. I proposed to discuss also the study of Stocco et al. because, to the best of my knowledge, this study is the only one, which suggests that these cells are primed by OA inflammation (with all the limitations of an in vitro study). Thus, these cells may not represent a useful source to treat osteoarthritis. Further studies deserve to be performed to clarify this topic.
